# Distractor inhibition by alpha oscillations is controlled by an indirect mechanism governed by goal-relevant information
Ole Jensen ✉

The role of alpha oscillations (8–13 Hz) in cognition is intensively investigated. While intracranial animal recordings demonstrate that alpha oscillations are associated with decreased neuronal excitability, it is been questioned whether alpha oscillations are under direct control from frontoparietal areas to suppress visual distractors. We here point to a revised mechanism in which alpha oscillations are controlled by an indirect mechanism governed by the load of goal-relevant information – a view compatible with perceptual load theory. We will outline how this framework can be further tested and discuss the consequences for network dynamics and resource allocation in the working brain.

## Alpha oscillations: From resting state to inhibition

Over the last decades, there has been a growing interest in the role of neuronal oscillations and how they support cognition[1]. In particular, there has been a strong focus on oscillations in the alpha band and how they might control the information flow in the working brain[2–7]. While it is clear that alpha oscillations are strongly modulated by attention, it is debated to what extent they reflect a mechanism of selective attention both in terms of distractor suppression[8] and gain control[9–11]. We will review empirical findings to discuss the *pros* and *cons* of this debate. We will then address a complementary account of how alpha oscillations are controlled, which can—at least partly—resolve the controversy. We promote the idea that the increase in alpha power in areas not engaged in a given task is governed by an indirect mechanism driven by the load of goal-relevant information[12]. Such a mechanism is compatible with perceptual load theory[13] and provides neuroscientific insight into a decade-old debate on whether distractors are indirectly suppressed[12,14,15].

Alpha oscillations were first discovered by Hans Berger in 1924[16]. The rhythm was detected in the EEG and has a frequency range from 8 to 13 Hz (note that slightly other rangers are used in the literature spanning from 7 to 13 Hz). The interest in those oscillations has since then come and gone[17]. For decades the interest in alpha oscillations was tempered by the view that they reflect a state of rest or idling[18] and, as such, are not directly involved in neuronal mechanisms supporting cognition. However, the finding that alpha oscillations can remain strong during working memory retention[19] and increase with working memory load[20] provoked a paradigm shift in which alpha oscillations were thought to reflect inhibition of visual input rather than a state of rest. It should be mentioned that the alpha power increase is observed when stimuli like letters, numbers or objects are maintained; however, if complex visuospatial information is maintained,

alpha power has been reported to decrease during working memory maintenance[21–24]. Nevertheless, the increase in the magnitude of alpha oscillations with working memory load when consonants, letters or faces are maintained is highly robust and has been reproduced in numerous studies[19,20,25,26]. These findings have resulted in the specific suggestion that the alpha activity inhibits the flow of visual input to protect working memory retention[4,27]. The inhibition notion can also explain why alpha power does not increase when complex visual information is maintained, as in this case, the maintenance likely relies on engaging a 'visuospatial sketchpad'[28]. Furthermore, the inhibition-by-alpha theory can also explain why alpha power increases during rest and eye-closed. In those cases, the alpha power increase serves to reduce the flow of visual input that might interfere with mind-wandering as well as reduce the metabolic demands of the occipital cortex when not engaged.

A second important finding is that alpha band oscillations are hemispherically lateralized with respect to spatial attention[29]. When for instance, a participant is cued to attend to the left, the alpha band activity in the contralateral right hemisphere is depressed while the alpha activity in the left ipsilateral hemisphere increases relatively. These types of studies have provided a workhorse paradigm allowing for investigating target and distractor processing in relation to brain oscillations, e.g. refs. 29–38. The hemispheric lateralization of alpha-band oscillations has also allowed well-controlled studies investigating the causal role of the alpha rhythm oscillations for spatial attention by directly manipulating the oscillations. These studies exploit the fact that alpha oscillations are easily entrained by brain stimulation[39–44] and visual flickers[45] in humans as well as by optogenetics in ferrets[46]. Specifically, studies using transcranial alternating current stimulation (tACS) as well as transcranial magnetic stimulation (TMS) have made it possible to entrain oscillations in one hemisphere and causally

Centre for Human Brain Health, School of Psychology, University of Birmingham, Birmingham B152TT, UK. ✉ e-mail: o.jensen@bham.ac.uk

substantiate their role in behaviour[37,47–51]. By entraining by visual stimulation it has been possible to drive one hemifield with a 10 Hz flicker while stimulating the other hemifield with a random stimulus train. This allowed for entraining the alpha oscillations selectively in one hemisphere resulting in a phasic modulation of detection ability[52]. Another line of studies has used hemispheric neurofeedback to investigate the causal role of the alpha rhythm. Specifically, these studies allowed for modulating the hemispheric asymmetry of the alpha oscillations by neurofeedback training. The testing after training revealed biases in spatial attention in the expected direction underscoring the causal role of the alpha oscillations[53,54] (but see ref. 55 for a critical perspective). In sum, hemispherically lateralized alpha-band oscillations modulated by spatial attention have made it possible to perform well-controlled experiments providing evidence for a causal role of alpha-band oscillations in the allocation of spatial attention.

A third development that has served to uncover the importance of alpha-band oscillations in relation to neuronal activity has been intracranial recordings in non-human primates. Before the year 2000, there were few reports on alpha band oscillations recorded in non-human primates which seemed at odds with the observations that alpha activity is the dominating rhythm in the human EEG. This discrepancy is likely explained by researchers focussing on quantifying the spiking of individual neurons rather than the local field potential. Also, electrodes are often placed in superficial or granular layers when single-unit intracranial recordings are performed in non-human primates. It is now clear from recordings in deeper cortical layers that there are strong alpha oscillations in the sub-granular layers[56–61]; but see ref. 62. Furthermore, it has also been established that the magnitude of the alpha band oscillations correlates inversely with both neuronal spiking and high-frequency activity[59,63–65]. Also, the phase of alpha oscillations modulates the neuronal spiking and gamma band activity[58,60,63,64,66–68]. These findings have resulted in the notion that the alpha oscillations observed in EEG recordings are due to neuronal firing being inhibited in a phasic manner at 8–13 Hz (Fig. 1). As more neurons are silenced in every cycle, the measured alpha power arising from the population activity increases (until sufficiently many neurons are depressed and the alpha power decreases). This hypothesis explains why the strongest measured signal from the human brain, the alpha rhythm, reflects inhibition[69]. Consistent with the inhibitory role of alpha oscillations, it is also important to note that simultaneous fMRI and EEG recordings have revealed that the BOLD response is inversely related to alpha power (see e.g. refs. 70–74). Given that alpha oscillations are reported to be generated in

deeper cortical layers[56,58,61] one could speculate that neurons in the deep cortical layers excise a pulsed suppressive drive to neurons in granular or super-granular layers. This hypothesis is consistent with optogenetic studies demonstrating that layer 5 cells exercise a suppressive drive on neurons in superficial layers[75]. However, the exact mechanism of suppression remains to be uncovered.

In sum, there is strong evidence that (1) alpha band oscillations play a role in orchestrating neuronal firing in the neocortex of humans and non-human primates, (2) alpha oscillations are associated with the inhibition of firing rates and (3) entrainment and neurofeedback studies speak to a causal role of alpha oscillations. It is, however debated whether alpha oscillations in general are under top-down control. A related issue is whether alpha oscillations are directly involved in gain control during the allocation of visual attention. We will discuss these topics and present a revised view in which alpha oscillations are indirectly controlled.

## Control of alpha oscillations by indirect mechanisms

It is debated whether alpha oscillations reflect the direct suppression of distractors in a top-down-driven manner[7,8,12,76]. We will here outline studies providing empirical data for and against the notion that alpha oscillations are under direct top-down control to suppress distractors. We then argue in favour of a complementary mechanism according to which distractor inhibition by alpha oscillations is driven by an indirect mechanism as a consequence of a load of goal-relevant information (Fig. 2). In line with the arguments above; we do argue that the indirect mechanism also relies on alpha oscillations being produced by pulsed inhibition which suppresses the information flow in a causal sense.

The notion that alpha oscillations reflect inhibition is motivated by alpha power remaining strong during working memory retention[19] as well as the increase in alpha power with working memory load[20,25,26] (unless complex visuospatial information is maintained). This alpha-inhibition hypothesis is supported by additional working memory studies in which anticipated distractors were presented during retention. Using MEG and EEG it was shown that alpha power increased before the onset of anticipated distractors[77–79]. Furthermore, the increase in alpha power just prior to the distractor onset predicted improved performance during working memory recall. Studies on spatial attention have also found evidence for alpha activity associated with distractor suppression. In a combined EEG and fMRI study, distracting visual representations were suppressed in the ventral stream as reflected by a decrease in the BOLD signal[36]. This object-specific

**Fig. 1 | A schematic plot demonstrating how alpha oscillations can be generated by pulsed inhibition. a** The raster represents the firing of a group of neurons. To the left, the firing rate of the neurons is high, and the firing is asynchronous. To the right, the firing is inhibited every ~100 ms, i.e. at ~10 Hz. **b** The population activity of the neurons above reflects the local field potential or scalp EEG. On the left, the neurons are firing strongly but asynchronous. As such, no modulation is observed in the population signal. The right oscillations are generated by pulsed inhibition silencing the neurons periodically. This scheme explains why an increase in alpha power is inversely correlated with the firing rate. Adapted from ref. 69.

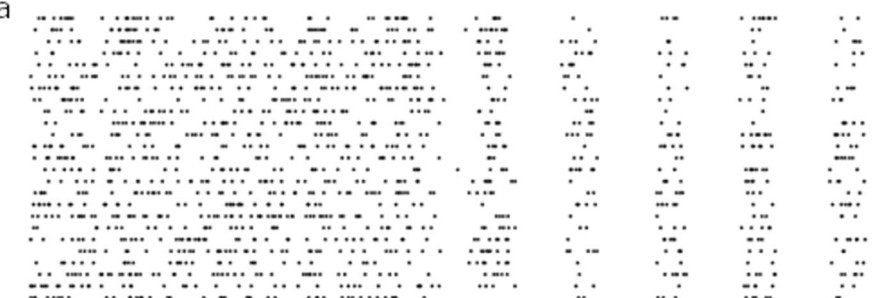
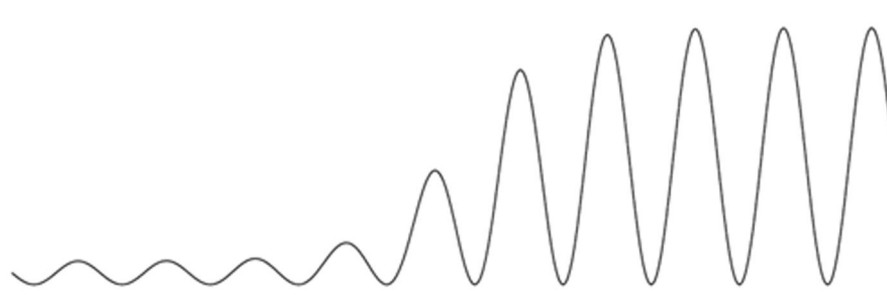

**Fig. 2 | Two mechanisms for modulating posterior alpha oscillations: direct feedback or indirect control. a** Direct top-down control of alpha oscillations driven by awareness or anticipation of, e.g. a right hemifield *Distractor* resulting in a power increase in the left posterior parietal cortex (PPC; around parieto-occipital sulcus). The alpha power increase serves to inhibit the feedforward flow of visual information associated with the *Distractor*. The control is exercised by regions in the dorsal attention network (blue regions); e.g. the frontal eye fields (FEFs) directly or via the intraparietal sulci (IPS). A subcortical route is also possible. **b** An indirect mechanism for the control of posterior alpha oscillations. The perceptual load of the left *Target* results in strong engagement of the right hemisphere dorsal attention network, thus promoting an alpha power increase in the left PPC (red regions) via lateral connections. This serves to reduce the flow of information associated with the *Distractor*. According to this revised framework, the alpha oscillations do not implement gain control by inhibiting early visual regions (green regions); rather, they serve to gate the information flow in the PPC (see also ref. 12).

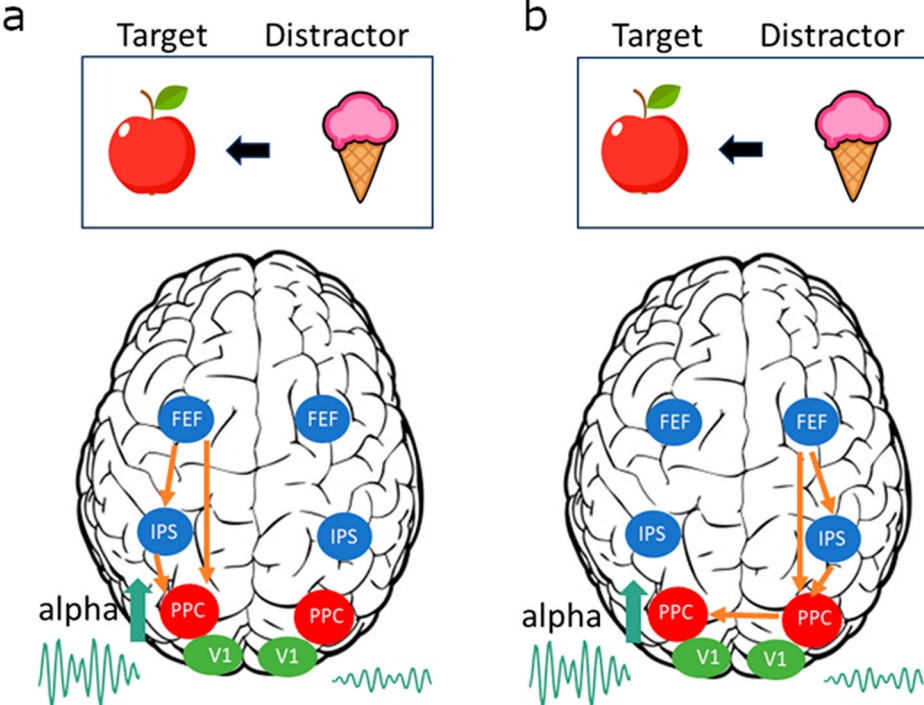

suppression correlated over trials with the alpha power increasing in the posterior hemisphere that was associated with the distractor. A spatial attention task in which targets were presented centrally and distractors in the left or right hemifield provided EEG evidence for an increase in alpha power associated with the distracting stimuli[34]. Similar findings were obtained in a statistical learning paradigm where the predictability of the distractor appearance was manipulated[80,81]. Similarly, a study where the perceptual load of both distractors and targets were manipulated provided support for stronger alpha power contralateral to strong versus weak distractors[82]. Using a paradigm with auditory targets presented centrally and distractors presented left or right, it was also found that alpha oscillations reflected suppression of contralaterally presented distractors[83]. Similar results were obtained using an auditory paradigm applying retro-cueing[84] as well as in a visual working memory task requiring hemifield-specific distractor suppression[32]. In sum, these studies suggest that alpha power associated with distractor inhibition can be under direct control; however, the general case for distractor inhibition by alpha oscillations has been questioned[8].

Several studies have, however, failed to find direct evidence for a top-down driven increase in alpha power associated with distractor suppression. For instance, in a study where respectively targets or distractors were cued, there was no evidence for alpha power reflecting the anticipated distractors[31]. A spatial working memory paradigm where targets in one hemifield were placed amongst either distracting or non-distracting stimuli failed to demonstrate an alpha power increase associated with the presence of distractors[33]. Finally, studies employing statistical learning paradigms manipulating the occurrence of distractors did not find an associated increase in alpha power[85,86].

In short, there is mixed evidence on whether alpha power reflected suppression of distractors in a direct top-down driven versus an indirect manner. It should be noted that the indirect control of the alpha oscillations is a consequence of a top-down drive engaging task-relevant regions, which then disengages task-irrelevant regions. As such, while alpha oscillations are undoubtedly associated with the inhibition of neuronal activity, the mechanisms controlling the alpha oscillations need to be better understood. In general, it has been suggested that distractor inhibition is a consequence of perceptual or cognitive load[13]. When target and distractor stimuli are

presented simultaneously, the distractor suppression might be an indirect consequence of the perceptual load of the targets rather than a direct suppression of the distractors[12]. Is there evidence for an indirect control mechanism for distractor inhibition reflected in the alpha band activity? This question was explicitly tested in a study where targets and distractors were presented in different hemifields[82]. The perceptual load was manipulated by adjusting the visibility of the target stimuli. Furthermore, the visibility of the distractors was similarly manipulated (Fig. 3). The core finding was that the perceptual load of the targets increased the alpha power associated with the distractors. This modulation in alpha power was also associated with behaviour: the increase in alpha power reduced the impact of the distractors on target detection as indexed by response times. This study also found evidence for an increase in alpha power associated with the visibility of the distractor; however, this effect was not predictive of the ability to suppress the impact of the distractor. From these findings, we conclude that a good part of the modulation of the alpha power is indirect and controlled by the perceptual load of the targets. Such a mechanism could serve the allocation of perceptual resources: In settings where the perceptual load is high and requires a lot of attentional resources, it is beneficial to suppress distracting stimuli. However, in settings with little perceptual load, it would make sense to also allocate resources in surplus to process stimuli of less relevance. This revised view of the role of alpha oscillations has great explanatory power and can partly resolve debates on distractor-related alpha power. For instance, in the study by Schroeder et al.[33], alpha power did not increase when distractors were introduced; it did, however, increase when the number of targets was increased in line with load theory. The increase in alpha power with working memory load[20,77,78] can also be explained by an increase in cognitive load, thus indirectly resulting in distractor suppression of the visual cortex. Also, the account of indirect modulation of distractor-related alpha power can explain the absence of alpha modulation in the statistical learning studies in which distractor appearance is manipulated[85,87]. It should, however, be mentioned that another EEG study on statistical learning reported enhanced power in parieto-occipital alpha oscillations contralateral to frequent distractor locations[81]. We conclude from the EEG and MEG studies that alpha oscillations and the subsequent distractor suppression can be controlled by both direct and indirect control mechanisms. While the indirect control

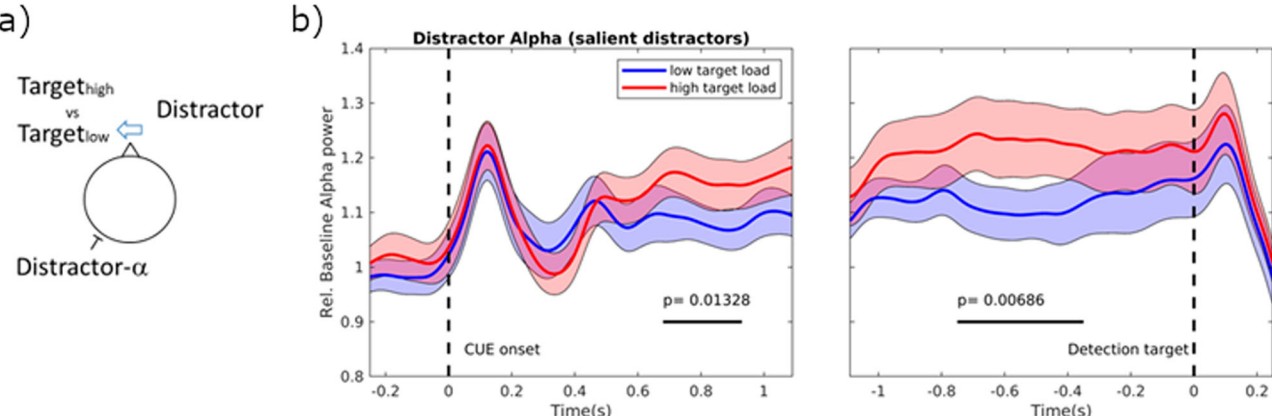

**Fig. 3 | The increase in distractor-related alpha power is explained by the perceptual target load. a** In this example, *Distractors* are presented to the right and *Targets* to the left. The perceptual load (visibility) of the *Targets* is manipulated. **b** The alpha power associated with the *Distractor* (i.e. the left hemisphere in this example) is stronger for high compared to low target loads. Reproduced from[82].

**Fig. 4 | When a 'virtual lesion' is provided to the frontal eye field (FEF) in one hemisphere by rTMS, the ability to modulate alpha power in the contralateral hemisphere is diminished.** This provides support for secondary, rather than direct, control of posterior alpha oscillations. **a** rTMS was applied to the left FEF, midline or right FEF in a blocked design. **b** After the rTMS was applied, the participants partook in a spatial attention task while the MEG was recorded. The *attentional modulation index* of alpha power (alpha power in a hemisphere when attention was allocated contralaterally versus ipsilaterally) is shown. **c** The ability to modulate the parieto-occipital alpha power was diminished in the hemisphere contralateral to the stimulated FEF. Reproduced from[88].

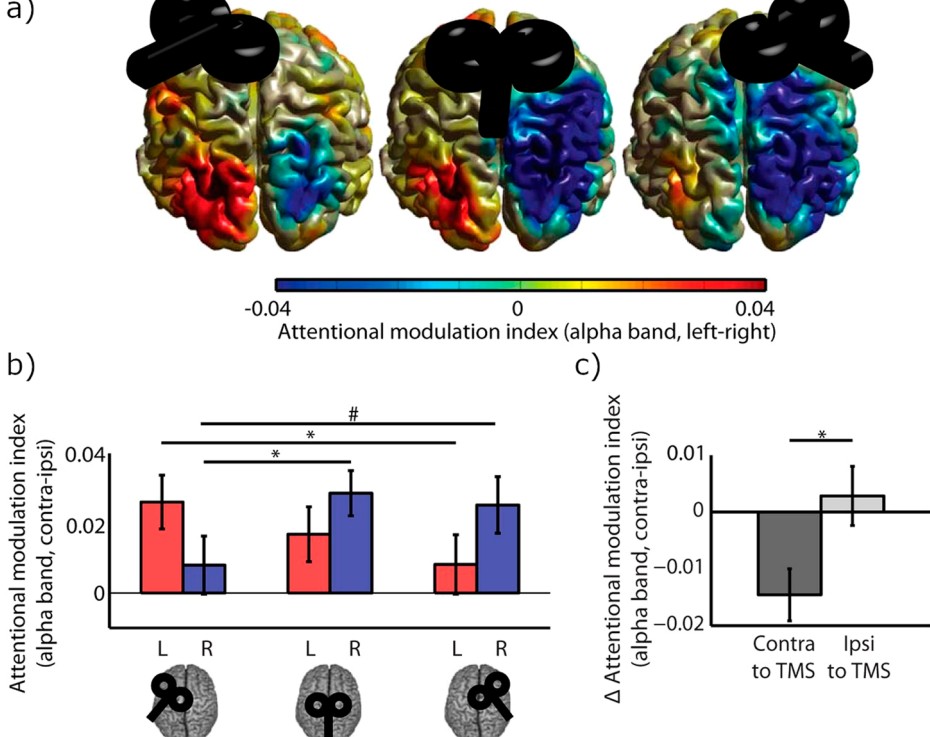

mechanisms appear to be dominant, further work is required to uncover in which settings indirect versus direct control mechanisms are at play. We do note that even though the increase in alpha power associated with distractor suppression is indirect, it is a secondary consequence of top-down control.

Can the direct versus indirect control mechanism of the alpha band oscillations be investigated in a causal sense? We here reinterpret findings from a study employing transcranial magnetic stimulation (TMS) in combination with MEG, which provides support for an indirect control mechanism (Fig. 4). In this study, repetitive TMS (rTMS) was applied to the left and right FEF as well as the midline (sham) in different blocks[88]. The rTMS protocol introduces a 'virtual lesion' in the FEF lasting up to 30 min. Immediately after the stimulation, the participants partook in a simple spatially cued attention task while the MEG was acquired. The key finding was that when rTMS was applied to the right FEF, the participants had a reduced ability to modulate the left parieto-occipital alpha oscillations when spatial attention was allocated. The reverse was found when the left FEF was stimulated. This finding is consistent with indirect control of the alpha oscillations: when one side of the dorsal attention network is impaired by rTMS, this reduces the ability to modulate the contralateral alpha oscillations (see also Fig. 2b). The rTMS findings are not consistent with direct top-down control of alpha oscillations. In that case, impairing one side of the dorsal control network should impact the ability to modulate alpha power in the same hemisphere (see Fig. 2a). It should also be mentioned that other brain stimulation studies have implicated the posterior intraparietal sulcus (IPS) in the control of posterior alpha-band oscillations[89–91]. In future work, it would be interesting to further uncover the roles of respectively the FEF and IPS in the control mechanism of posterior alpha oscillations. It would also be important to uncover the specific pathways. When, for instance, the

right FEF is impaired, does it reduce the ability to modulate the left posterior alpha directly via transcallosal connection or indirectly via transcallosal connections from the IPS?

## Do alpha oscillations do not implement sensory gain control?

Numerous studies have demonstrated that the allocation of spatial attention modulates the response gain in early visual areas. This has resulted in the hypothesis that gain control in sensory regions is implemented by alpha oscillations via disinhibition, as explicitly proposed in ref. 92. However, a set of studies based on frequency tagging has brought into question whether this holds in general. When employing frequency tagging, a visual object is flickered and the neuronal response is measured with EEG or MEG. The tagging response reflects neuronal excitability in early visual regions and is modulated by spatial attention. Using both slow (14–20 Hz) and fast (60–70 Hz) frequency tagging, it was demonstrated that the flicker response is not modulated by the magnitude of the alpha oscillations over trials[9–11]. These findings cannot be reconciled with the hypothesis that alpha oscillations implement gain control in early visual areas. Importantly, a MEG study localized the frequency tagging modulation in early visual regions, whereas the attention modulation of alpha power was observed around the parieto-occipital sulcus[11]. We, therefore, suggested that alpha oscillations reflect the gating of visual information slightly downstream to the visual system. This notion is consistent with other studies that have demonstrated that the decrease in alpha power is predictive of long-term memory encoding as the gating allows for more efficient encoding in hippocampal memory areas[93]. A study entraining oscillations in the posterior parietal cortex using optogenetics in ferrets concluded that alpha oscillations operate in a phase-dependent manner gating the responsiveness to visual input[46]. As such, while alpha oscillations may not reflect top-down controlled sensory processing in early visual areas, they are involved in resource allocation in cortical association areas. This conclusion does, however not exclude that alpha oscillations can be generated and modulate neuronal firing in early visual areas. Indeed, alpha oscillations have been detected in early visual regions using intracranial recording in non-human primates[56–58,60]. Recordings in non-human primates have demonstrated an inverse relationship between the magnitude alpha oscillations in V1 and power in the gamma band[56,58]. Also, the C1 response, an ERP in humans thought to be generated in the primary visual cortex, correlates negatively with pre-stimulus alpha power while it is also modulated by the phase of the alpha oscillations[94,95].

When visual representations were investigated by multivariate pattern analysis applied to the BOLD signal, it was shown that a decrease in posterior alpha oscillations predicted an increase in the fidelity of visual representations in the occipital cortex[96]. The conjunction of these studies suggests that while the alpha oscillations generated in early visual regions might inhibit neuronal firing, they are not under direct top-down control in spatial attention tasks, as is the case for alpha oscillation generated around parieto-occipital sulcus.

## How general is the role of alpha oscillations for resource allocation?

Most of the studies reviewed above have focussed on the role of alpha oscillations during attention and working memory tasks. To what extent might the inhibition by alpha oscillations play a role in resource allocation for other cognitive operations? Indeed, one would be hard-pressed to find a cognitive task in which no modulations in the alpha band are observed. It would therefore be of great interest to uncover if the functional role of the alpha oscillations generalizes beyond parieto-occipital areas. It is well-established that oscillatory activity in the alpha band is also generated in sensorimotor areas[64,97–100]. Indeed, these alpha oscillations are depressed both during motor and somatosensory tasks: when somatosensory attention is allocated to one hand, the alpha oscillations decrease in the contralateral sensorimotor cortex while they increase in the other hemisphere[98,99]. Importantly, the ipsilateral alpha

power increase is related to the suppression of somatosensory distractors[101]. It is interesting to note that, unlike in the visual system, the alpha oscillations in the sensorimotor system do seem to relate to gain control. For instance, it was demonstrated that motor-evoked potentials induced by TMS over the motor cortex are reduced with increased alpha power over sensorimotor cortex[102].

Alpha oscillations have also been related to motor action. For instance, it has been demonstrated that parietal alpha oscillations are modulated by the planning of eye movements as they decrease in the hemisphere contralateral to the direction of the saccade[103]. As the alpha modulation emerges during the planning of the saccade, proprioceptive contribution from the somatosensory system can be excluded. These findings are consistent with more recent work demonstrating that micro-saccades and alpha oscillations are co-modulated in spatial attention tasks[104]. Also reaches have been found to modulate the posterior alpha band activity—interesting the lateralization was coded in gazed centred manner[105]. As such, there is ample evidence for the generation and task-specific modulation of alpha-band oscillations in the motor system. As for the visual domain, the alpha oscillations decrease in sensorimotor as well as parietal regions associated with engagement while they increase in areas not required for the task.

Alpha oscillations might also be directly involved in coordinating and modulating neuronal activity in some areas of the prefrontal cortex. The left-hemispheric dominance of the human language system offers an opportunity to study the regional specificity of alpha oscillations associated with reading. Alpha oscillations are suppressed in the left prefrontal related to Broca's area and temporal language areas when sentences are presented visually word-by-word[106,107]. Importantly, the alpha band activity in left language areas (around Broca's and Wernicke's areas) decreased when words were presented that were incongruent with the sentence context[107]. Executive control is another important topic. Using intracranial multielectrode recordings from the pre-frontal cortex in non-human primates it was demonstrated that distributed neuronal activity in the alpha band supports the de-selection of rules to be applied in a visual task[108]. Other studies on non-human primate recordings have demonstrated that the trajectories of covert attention (termed 'attentional saccades') can be decoded from a population of neurons in the frontal eye field[109]. This spatial exploration was dominated by a 7–12 Hz rhythm. In sum, while more work needs to be done, these findings are consistent with alpha oscillations also playing an important role in coordinating neuronal processing and allocating computational resources in the prefrontal cortex. As several prefrontal subregions are connected to posterior and subcortical regions., it remains to be uncovered how the allocation of executive resources in the prefrontal cortex by alpha oscillations interacts with the allocation of computational resources in posterior brain regions.

Alpha oscillations have also been found to be modulated in auditory tasks. When spatial attention is allocated to the left or right auditory modality, the typical hemispheric lateralization is observed in the alpha band[83,110–113]. It does, however, remain to be clarified if and when the alpha oscillations reported in the EEG and MEG studies primarily are produced in auditory areas as observed using intracranial recordings[114]. Alternatively, they might be generated in parietal regions associated with supramodal allocation of spatial attention[115,116]. Cross-modal attention tasks involving the visual and auditory modalities also produce robust modulations of alpha band activity. There is no doubt that posterior alpha oscillations increase when attention is allocated to the auditory modality[117–119], it remains more of a question if alpha oscillations increase in the auditory cortex when attention is allocated to the visual modality. Nevertheless, some studies show alpha oscillations in the supramarginal gyrus of the temporal cortex increase when attention is allocated to the visual versus the auditory modality[7,118]. In sum, all these findings are consistent with alpha oscillations reflecting the allocation of resources in both frontal, temporal and parietal regions. Nevertheless, the functional role of alpha oscillations in these regions remains to be further uncovered.

## Network interactions revealed by alpha band activity

The brain should be studied as a network in the context of cognitive tasks being performed. We will here summarize some of the key aspects associated with the control of alpha oscillations from a network perspective albeit we refer to other review papers for a more comprehensive account[5,120–124]. As pointed out earlier, the frontal eye field (FEF) seems to exercise control of posterior alpha oscillations in attention tasks. Measures applying spectral Granger analysis quantifying phase-to-phase interactions in the alpha band of MEG data from participants performing a spatial attention task point to top-down control emerging from the FEF[125]. These findings are complemented by TMS studies demonstrating that the ability to modulate alpha oscillations is impaired when the FEF is perturbed[88,91,126]. Using a similar approach also the intra-parietal sulcus has been implicated[89,91]. These interactions are likely to be mediated, at least in part, by the superior longitudinal fasciculus, as demonstrated by an MEG study combined with diffusion tensor imaging (DTI)[127]. In the context of the discussion on whether alpha oscillations are under direct or indirect top-down control, it remains to be further uncovered whether the FEF exercises a direct or indirect modulation of the alpha oscillations. When considering the control and network interactions in the alpha band, subcortical regions are likely to play an important role[128–130]. From recordings in non-human primates, it was demonstrated that alpha oscillations in the pulvinar are phase-synchronized with neocortical oscillations[131]. Importantly, the pulvinar served to synchronize two extra-striate visual regions in order to support the information flow in a spatial attention task. Also, areas in the basal ganglia have been implicated in the control of posterior alpha oscillations[132,133]. In sum, while some of the key regions controlling alpha oscillations have been identified, there is a lot to uncover in terms of the specific pathways and mechanisms involved.

## Future directions

The considerations addressed above call for adjusting the theories of the functional role of alpha-band oscillations. While there is no doubt that alpha oscillations are inhibitory and correlated with a reduction in neuronal excitability, they might not be under direct top-down control in general. The view of alpha oscillations being under indirect control is consistent with them not implementing gain control in early sensory regions. Rather we promote the idea that alpha oscillations are modulated by an indirect control mechanism[12,82], in which the load of goal-oriented processing results in the control of the alpha power increases in task-irrelevant regions. This framework is compatible with perceptual load theory[13]. The revised framework on the inhibition exercised by alpha oscillations results in a set of predictions and open questions.

As proposed in Fig. 2b, the engagement of areas in the dorsal attention network in one hemisphere promotes the increase of alpha power and, thus, a decrease of neuronal excitability in the other hemisphere. This suggests a lateral competitive mechanism between the two hemispheres, which is most likely mediated by the corpus callosum or subcortical regions. However, is this mechanism operating between regions at the same level in the hierarchy (e.g. left versus right PPC) or can it be a consequence of e.g. the left FEF acting on the right PPC? This question could be answered by extending the rTMS approach of Marshall et al.[88], Beyond applying rTMS to the FEF, one could also target the IPS, FEF and V1/2 in one hemisphere, after which the participants then partake in a cued spatial attention task. From MEG or EEG data acquired after the rTMS intervention, one would be able to uncover at which level in the hierarchy of the dorsal attention network the cross-hemispheric interactions are implemented. A related question pertains to the specific physiological mechanisms by which the competitive interactions are implemented. Connectivity across the corpus callosum is typically mediated by excitatory connections. However, these connections might serve to engage inhibitory interneurons pulsing at ~10 Hz, thus generating the alpha oscillations by rhythmically silencing excitatory neurons (see Fig. 1). This engagement could be via direct synaptic interactions or a neuro-modulatory drive; however, the specific mechanism warrants further investigation. It would be of great interest to test these possibilities using

animal electrophysiology paired with optogenetic approaches. Such tools are currently being developed and applied to study alpha oscillations in mice and ferrets[46,134].

Might alpha oscillations be involved in implementing inhibition between competing modalities? For instance, when competing auditory and visual stimuli are presented, the alpha power has been found to increase in the extra-sensory regions associated with the distracting modality[118] (see also refs. [112,114]). Likewise, parieto-occipital alpha oscillations increase in tasks requiring attention allocated to the somatosensory modality[98]. Competition between the dorsal and ventral stream might also be reflected in the alpha band. In a working memory study in which participants had to maintain either the identity or orientation of a face (engaging, respectively, the ventral or dorsal stream), alpha oscillations increased in the dorsal stream when face identity was maintained[135]. These findings are complemented by a related EEG study also manipulating feature attention in which participants had to attend to either colour or direction of motion in random-dot kinematograms. The alpha power increased in the dorsal or ventral regions when the irrelevant features were respectively colour or motion[136]. These cross-modal and feature-specific studies did not rely on transcallosal competition, and it would be interesting to uncover if the alpha power increase associated with the distractor-related modality or feature is under direct or indirect control. This could again be addressed with rTMS protocols. In general, more research is needed to uncover which processes and, thus, brain regions are competing for resources. This question is best answered by MEG recordings from a large range of paradigms to uncover where and when alpha oscillations increase when the load of cognitive demands increases in regions not required for a given task.

The framework in which alpha power increases in task-irrelevant regions when task-relevant regions are engaged points to competitive interactions. There is clearly an interhemispheric competition when targets and distractors are presented in different hemifields. This cross-hemispheric competition applies to the visual[29,30], auditory[83,112] and somatosensory domains[101]. What about competition on smaller spatial scales, e.g. competition within a visual hemifield? Using intracranial recordings, it has been suggested that centre-surround competition is implemented by alpha oscillations[137]. This interpretation was later adjusted based on findings in another intracranial study. It was found that alpha oscillations in the visual cortex indeed decrease when a stimulus is presented in the respective receptive field. However, the suppression in the alpha band extends beyond the size of the classical respective field and the alpha oscillation then increases at even further eccentricities[138]. These effects operating at spatial scales within a hemifield make it possible that alpha oscillations reflect the suppression associated with bias competition[139]. To investigate the role of alpha oscillations in biased competition is best done using intracranial recordings in humans or non-human primates as MEG or EEG recordings cannot provide the required spatial resolution. It would also be interesting to investigate if the principles of biased competition[139,140] and normalization theory[141] in a broader sense apply to interhemispheric competition and it related to suppression by alpha oscillations.

Finally, the specificity of the alpha band range deserves to be further uncovered. Often modulations of oscillations in the beta band (13–30 Hz) reflect engagement and disengagement, in particular in the motor system; however, beta modulations are also observed in posterior regions[142]. While oscillations of theta oscillations are defined in the 4–8 Hz range in the human EEG, the hippocampal theta rhythm spans 4–12 Hz[143]. Interestingly the hippocampal theta oscillations are also a consequence of pulses of GABAergic inhibition originating from the medial septum[144]. This raised the possibility that theta and alpha oscillations play functionally similar roles, resulting in the hypothesis that communications between the visual system and the hippocampus might be mediated by phase-synchronization in the theta-alpha band[145].

## Conclusion

In this review, we have made a case for the modulation of alpha oscillations being due to an indirect control mechanism. According to this mechanism,

it is a load of goal-driven information that promotes the alpha power increase in regions processing potentially distracting task-irrelevant information. According to this mechanism, alpha power is not in general under direct top-down control, and it, therefore, explains the studies failing to find that alpha oscillations exercising gain control in early visual regions as well as why some studies do not find an increase in alpha power with respect to distractor load. However, it is important to note that the mechanism does not preclude alpha power from also being modulated by direct top-down control. The stage is now set to further uncover which regions and neuronal mechanisms are involved in the indirect control and generation of alpha oscillations. This is best done using interventional approaches such as rTMS in which the regions suspected to exercise the control are temporarily disengaged, followed by EEG or MEG recordings in which the local impact on neuronal excitability or behaviour of the task-irrelevant alpha oscillations are characterized. Furthermore, we call for electrophysiological investigations in animals combined with optogenetic tools in order to uncover the specific mechanisms associated with the generation and control of alpha oscillations.

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

## Acknowledgements

The work was supported by funding from a Wellcome Trust Discovery Award (grant number 227420) and by the NIHR Oxford Health Biomedical Research Centre (NIHR203316). The views expressed are those of the author(s) and not necessarily those of the NIHR or the Department of Health and Social Care. The funders had no role in the preparation of the manuscript or decision to publish.

## Competing interests

The author declares no competing interests.
