## [Peer Review File · Communications Psychology]

17th Jan 24

Dear Ole,

Thank you for your patience during the peer-review process. Your manuscript titled "Gating by alpha band inhibition revised: a case for an indirect control mechanism" has now been seen by 3 reviewers, and I include their comments at the end of this message.

The reviewers share our enthusiasm for your work. However, they also mention a number of concerns. We are very interested in the possibility of publishing your manuscript in *Communications Psychology*, but would like to consider your response to these concerns in the form of a revised manuscript before we make a decision on publication.

As you will see, there are some elements of disagreement and some requests for clarification in the reviews. We ask you to address each concern in your revision.

In sum, we invite you to revise your manuscript taking into account all reviewer and editor comments.

EDITORIAL POLICIES AND FORMATTING

You will find a complete list of formatting requirements following this link:

<https://www.nature.com/documents/commsj-style-formatting-checklist-review-perspective.pdf>

Please use the checklist to prepare your manuscript for resubmission.

*** TRANSPARENT PEER REVIEW:** *Communications Psychology* uses a transparent peer review system. This means that we publish the editorial decision letters including Reviewers' comments to the authors and the author rebuttal letters online as a supplementary peer review file. We publish these records for all accepted manuscripts. However, on author request, confidential information and data can be removed from the published reviewer reports and rebuttal letters prior to publication. If your manuscript has been previously reviewed at another journal, those Reviewers' comments would not form part of the published peer review file.

If you have any questions about any of our policies or formatting, please don't hesitate to contact me.

Please use the following link to submit your revised manuscript and a point-by-point response to the referees' comments (which should be in a separate document to any cover letter):

[link redacted]

****** This url links to your confidential home page and associated information about manuscripts you may have submitted or be reviewing for us. If you wish to forward this email to co-authors, please delete the link to your homepage first ******

We hope to receive your revised paper within 12 weeks; please let us know if you aren't able to

submit it within this time so that we can discuss how best to proceed. If we don't hear from you, and the revision process takes significantly longer, we may close your file.

Please do not hesitate to contact me if you have any questions or would like to discuss these revisions further. We look forward to seeing the revised manuscript and thank you for the opportunity to review your work.

Best wishes,
Marike

Marike Schiffer, PhD
Chief Editor
Communications Psychology

REVIEWERS' EXPERTISE:

All reviews have a background in cognitive neuroscience and electrophysiology.

REVIEWERS' COMMENTS:

Reviewer #1 (Remarks to the Author):

This Perspective presents the idea that alpha oscillations index an indirect form of inhibition of irrelevant or distracting information, particularly in vision, where the inhibition results from the perceptual load associated with processing of task-relevant target stimuli. The review mostly focuses on M/EEG results but also draws on findings from fMRI and animal intracranial studies.

I'm writing this review as someone who has worked on alpha and attention but who is not currently immersed in that literature. As such I found the Perspective interesting and well-written. But I also had some concerns, confusions and questions about the manuscript in its current form.

The aim and scope of the Perspective could be spelled out more clearly. At times the focus seems to be answering the question of whether alpha reflects direct inhibition or indirect effects of target processing load. However, without more theoretical context, this seems like a niche question that is best answered in the discussions of empirical papers of the kind reviewed on pp.4-5 rather than needing separate extended discussion of this kind of Perspective. At other times, the Perspective seems to have the more ambitious aim of situating findings about EEG alpha in a wider understanding of attentional networks and theories of attention. However, some of the main claims could be stated more clearly, and overall the Perspective could make clearer which are the key claims and which are more speculative (less critical) claims:

- There is confusion about whether alpha is the source, mechanism or consequence of attention/inhibition. For example, whereas Line 108 talks about "whether alpha oscillations reflect the suppression of distractors in a top-down-driven manner", line 110 considers whether "alpha oscillations perform top-down controlled distractor suppression", and line 301 talks about "control of alpha oscillations". This issue seems fundamental to the Perspective and so needs to be explained more clearly and consistently.
- Further confusion is added by the statement on lines 154-155 that "there is mixed evidence on

whether alpha power reflected suppression of distractors in a top-down driven versus an indirect manner". I don't see there being a contradiction between "top-down" and "indirect". As shown in Figure 2B, even if alpha/inhibition is indirectly caused by target processing, that processing is itself dependent on top-down control by the dorsal attention network.

- The debate/theory illustrated in Figure 2 could also be clarified. What is the difference between the solid and dotted arrows? Why is the mechanism in (B) characterized as "indirect" when there are arrows from right FEF and right IPS directly to left PPC? What is signified by the color of the brain area labels? In particular, I wasn't clear what is the proposed function of PPC (red) that explains or relates to its role as the site and/or source of alpha power modulations.

- I wasn't clear if the pulsed inhibition mechanism shown in Figure 1 is intended to be the proposed mechanism or just an illustrative example of how alpha could be generated (without committing to this particular mechanism). Is there research exploring how this kind of inhibition could be generated by target processing or perceptual load, and why it has the properties of being pulsed and with alpha frequency?

- The discussion of alpha in PFC in the paragraph starting on Line 272 adds further complexity. What is controlling these alpha oscillations, when many theories of cognitive control would characterize PFC as the source of this kind of control?

- I was surprised by the discussion on Lines 330-335 of a "push-pull" interhemispheric interaction. This discussion implies there is something fundamental to attention about interhemispheric interactions, but I think of the left vs. right hemisphere contrast as an experimental convenience to help with the low spatial resolution of EEG rather a fundamental claim about mechanisms of attention and alpha. More localized alpha effects have been observed, including in the author's own work, and if alpha power changes relate to a general mechanism for attention then they shouldn't they be observed even if target and distractors are presented in the same hemifield?

- The focus on the perceptual load theory of attention may limit the impact of the Perspective, if readers interpret the claims made as being specific to this theory. Although it is still cited reasonably widely, I don't see perceptual load theory as a major driver of current research, particularly in cognitive and systems neuroscience research that this Perspective aims to integrate. In contrast, I was struck that the Perspective only mentions biased competition in the final sentence before the conclusion, when this theory seems much more influential on and relevant to current neuroscience approaches to attention.

Minor line-by-line comments

Line 22. Check reference 15. No author info etc.

Lines 178-180. Wouldn't cognitive load reduce rather than increase inhibition, following the logic and results of studies like De Fockert et al. (2001)?

Line 272. "modulation" should be "modulating".

Line 370. "but SEE".

Reviewer #2 (Remarks to the Author):

The present review introduces a novel perspective on the indirect control mechanism governing alpha-based suppression in memory and attention, offering a fantastic and timely contribution to the field. The well-organized evidence supporting this concept aligns with my personal viewpoint. I have several suggestions to enhance the paper.

1, Concerning the term “top-down”, an ongoing debate in the field of attention questions its definition (see Theeuwes, 2019, JoC and the commentaries to this paper). Generally, the argument revolves around the assertion that top-down control is intentional, effortful and goal-directed. However, the term “top-down direct control” adopted in the present review may not imply goal-directed control; instead, it might only refer to the direct link between alpha oscillation and suppression. Thus, using “direct and indirect control” could suffice for readers to grasp the concept.

2, In line 35. “The increase in the magnitude of alpha oscillations with working memory load is highly robust and has been reproduced in numerous studies¹⁷⁻²⁰”. While I appreciate these results, lots of experiments have shown that alpha power decreases with an increase in working memory load, (e.g., Fukuda & Woodman, 2017; Liu et al., 2022; Wang et al., 2020; for a review see Woodman et al., 2022). It is crucial to reconcile these findings with the current indirect control account. Similarly, in line 180, “Also, the account of indirect modulation of distractor related alpha power can explain the absence of alpha modulation in the statistical learning studies in which distractor appearance is manipulated 71,73 (but see 65,66)”. I thought the study from 66 is consistent with alpha suppression account (Woodman et al., 2022).

Minor

1, Another study delves into the relationship between perceptual load and alpha, which could provide valuable insights (Wang et al., 2020).

2, The definition of alpha varies across existing literature, ranging from 8-12Hz, 8-13Hz, or 7-12Hz, etc.. Could the author provide a note/rule for future studies to follow?

3, A well-defined figure describing the boundary between direct and indirect control would assist readers in understanding how alpha oscillation operates in suppression.

Fukuda, K., & Woodman, G. F. (2017). Visual working memory buffers information retrieved from visual long-term memory. *Proceedings of the National Academy of Sciences*, 201617874.

<https://doi.org/10.1073/pnas.1617874114>

Liu, B., Li, X., Theeuwes, J., & Wang, B. (2022). Long-term memory retrieval bypasses working memory. *NeuroImage*, 261, 119513. <https://doi.org/10.1016/j.neuroimage.2022.119513>

Wang, S., Megla, E. E., & Woodman, G. F. (2020). Stimulus-induced Alpha Suppression Tracks the Difficulty of Attentional Selection, Not Visual Working Memory Storage. *Journal of Cognitive Neuroscience*, 1–27. https://doi.org/10.1162/jocn_a_01637

Woodman, G. F., Wang, S., Sutterer, D. W., Reinhart, R. M. G., & Fukuda, K. (2022). Alpha suppression indexes a spotlight of visual-spatial attention that can shine on both perceptual and memory representations. *Psychonomic Bulletin & Review*, 29(3), 681–698.

<https://doi.org/10.3758/s13423-021-02034-4>

Reviewer #3 (Remarks to the Author):

This is a really beautiful review article advocating the idea of posterior alpha activity reflecting a gating (instead of an inhibition) mechanism - a gating mechanism that is indirectly controlled via e.g. FEF. It is convincingly demonstrated that this gating is driven by cognitive load.

I absolutely enjoyed this read, and I must admit, it also got me a bit excited about alpha activity

again. Although there are quite a few similar articles out already, this review paper adds a novel component to the topic. The manuscript is nicely written and easy to follow, making it suitable for a broad readership.

I have a few minor points that the author might want to address in a revision.

1) The paragraph around line 55 describes approaches how to drive alpha activity using magnetic, electric and photic stimulation. It might be considered to also add neurofeedback-training here. I think the paper by Bagherzadeh et al. (2020) in *Neuron* would very nicely tie in here.

2) I am not so super convinced by the "how general is the role of alpha oscillations..." chapter. Sure, there is modulation of alpha activity in sensorimotor areas as well. But in terms of the proposed mechanism, is it really comparable with visual alpha? Hummel et al. (2002) *Brain* showed quite nicely that intentional action inhibition is associated with sensorimotor alpha activity. High alpha activity then also led to reduced cortico-spinal excitability (TMS induced MEPs). So this would actually be an example of alpha amplitude reflecting gain control in early sensory (or here motor) areas, wouldn't it. To be honest, I think the manuscript could do very well without the general role of alpha chapter.

3) Instead, I would find it exciting to have something in the paper on how frequency specific this effect really is. There are sometimes effects very similar to those at alpha frequency reported for beta, sometimes even theta. So, in how far is this cognitive load dependent gating mechanism specific for alpha and cannot be applied to other frequencies? I am aware that this might go beyond this manuscript's scope. So I will let it to the author whether they want to include something on it.

4) around line 307 connectivity between FEF and parietal cortex and alpha is discussed. There is a paper by Sauseng et al. (2011) *Front Psychol* where FEF was TMS-ed and the effect on parietal alpha and fronto-parietal interaction was investigated during visual attention cueing. Maybe this paper could be interesting here as well.

5) line 370 seems to miss a reference: ...different hemifields (but SEE)... see what?

6) reference 15 is not complete.

Dear Marike

Thanks for reviewing the manuscripts. I found the referee's comments highly constructive, and they have helped to significantly improve the manuscript. Below please see the detailed response to the concerns. Responses are in blue.

Best wishes,

Ole

Reviewer #1 (Remarks to the Author):

The aim and scope of the Perspective could be spelled out more clearly. At times the focus seems to be answering the question of whether alpha reflects direct inhibition or indirect effects of target processing load. However, without more theoretical context, this seems like a niche question that is best answered in the discussions of empirical papers of the kind reviewed on pp.4-5 rather than needing separate extended discussion of this kind of Perspective.

I have now added papers and a general discussion on the secondary suppression mechanisms in the context of biased competition and normalisation theory. I have also made adjustments throughout the text where appropriate to sharpen the perspective.

Line 28

“Such a mechanism is compatible with perceptual load theory¹³ and provides neuroscientific insight into a decade-old debate on whether distractors are indirectly suppressed^{12,14,15}”
Line 442

“It would also be interesting to investigate if the principles of biased competition^{144,145} and normalization theory¹⁴⁶ in a broader sense apply to interhemispheric competition and its related to suppression by alpha oscillations.”

At other times, the Perspective seems to have the more ambitious aim of situating findings about EEG alpha in a wider understanding of attentional networks and theories of attention.

However, some of the main claims could be stated more clearly, and overall the Perspective could make clearer which are the key claims and which are more speculative (less critical) claims:
- There is confusion about whether alpha is the source, mechanism or consequence of attention/inhibition. For example, whereas Line 108 talks about “whether alpha oscillations reflect the suppression of distractors in a top-down-driven manner”, line 110 considers whether “alpha oscillations perform top-down controlled distractor suppression”, and line 301 talks about “control of alpha oscillations”. This issue seems fundamental to the Perspective and so needs to be explained more clearly and consistently.

Indeed the text could have been clearer. I updated several places e.g.

On line 129 (~line 108 before) we now clarify

“It is debated whether alpha oscillations reflect the direct suppression of distractors in a top-down-driven manner^{7,8,12,76}. We will here outline studies providing empirical data for and against the notion that alpha oscillations are under direct top-down control to suppress distractors. We then argue in favour of a complementary mechanism according to which distractor inhibition by alpha oscillations is driven by an indirect mechanism as a consequence of the load of goal-relevant information (Figure 2). In line with the arguments above, we do argue that the indirect mechanism also relies on alpha oscillations being produced by pulsed inhibition which suppresses the information flow in a causal sense. “

And on Line 349

“In the context of the discussion on whether alpha oscillations are under direct or indirect top-down control, it remains to be further uncovered whether the FEF exercises a direct or indirect modulation of the alpha oscillations.”

Online 74 we clarify:

“Another line of studies has used hemispheric neurofeedback to investigate the causal role of the alpha rhythm. Specifically, these studies allowed for modulating the hemispheric asymmetry of the alpha oscillations by neurofeedback training. The testing after training revealed biases in spatial attention in the expected direction underscoring the causal role of the alpha oscillations^{54,55} (but see⁵⁶ for a critical perspective). In sum, hemispherically lateralized alpha-band oscillations modulated by spatial attention have made it possible to perform well-controlled providing evidence for a causal role of alpha-band oscillations in the allocation of spatial attention. “

Also, Figure 2 and caption has been updated.

- Further confusion is added by the statement on lines 154-155 that “there is mixed evidence on whether alpha power reflected suppression of distractors in a top-down driven versus an indirect manner”. I don’t see there being a contradiction between “top-down” and “indirect”. As shown in Figure 2B, even if alpha/inhibition is indirectly caused by target processing, that processing is itself dependent on top-down control by the dorsal attention network.

On line 180 we now clarify

“It should be noted that the indirect control of the alpha oscillations is a consequence of a top-down drive engaging task-relevant regions which then disengages task-irrelevant regions.”

And on Line 349

“In the context of the discussion on whether alpha oscillations are under direct or indirect top-down control, it remains to be further uncovered whether the FEF exercises a direct or indirect modulation of the alpha oscillations.”

- The debate/theory illustrated in Figure 2 could also be clarified. What is the difference between the solid and dotted arrows? Why is the mechanism in (B) characterized as “indirect” when there are arrows from right FEF and right IPS directly to left PPC? What is signified by the color of the brain area labels? In particular, I wasn’t clear what is the proposed function of PPC (red) that explains or relates to its role as the site and/or source of alpha power modulations.

Indeed the Fig 2 could be improved and better explained. I have now updated the figure as well as the caption.

- I wasn't clear if the pulsed inhibition mechanism shown in Figure 1 is intended to be the proposed mechanism or just an illustrative example of how alpha could be generated (without committing to this particular mechanism). Is there research exploring how this kind of inhibition could be generated by target processing or perceptual load, and why it has the properties of being pulsed and with alpha frequency?

Excellent point. On line 100 we now clarify

"Given that alpha oscillations are reported to be generated in deeper cortical layers^{57,59,62} one could speculate that neurons in the deep cortical layers excise a pulsed suppressive drive to neurons in granular or super-granular layers. This hypothesis is consistent with optogenetic studies demonstrating that layer 5 cells exercise a suppressive drive on neurons in superficial layers⁷⁵. However, the exact mechanism of suppression remains to be uncovered."

- The discussion of alpha in PFC in the paragraph starting on Line 272 adds further complexity. What is controlling these alpha oscillations, when many theories of cognitive control would characterize PFC as the source of this kind of control?

This is a good point. On line 318 I now clarify:

"In sum, while more work needs to be done, these findings are consistent with alpha oscillations also playing an important role in coordinating neuronal processing and allocating computational resources in the prefrontal cortex. As several prefrontal subregions are connected to posterior and subcortical regions., it remains to be uncovered how the allocation of executive resources in prefrontal cortex by alpha oscillations interacts with the allocation of computational resources in posterior brain regions."

- I was surprised by the discussion on Lines 330-335 of a "push-pull" interhemispheric interaction. This discussion implies there is something fundamental to attention about interhemispheric interactions, but I think of the left vs. right hemisphere contrast as an experimental convenience to help with the low spatial resolution of EEG rather a fundamental claim about mechanisms of attention and alpha. More localized alpha effects have been observed, including in the author's own work, and if alpha power changes relate to a general mechanism for attention then they shouldn't they be observed even if target and distractors are presented in the same hemifield?

Yes, I can see why this is confusing. I have now removed the push-pull terminology and refer to competitive interactions.

On line 373 I now write:

"This suggests a lateral competitive mechanism between the two hemispheres which is most likely mediated by the corpus callosum or subcortical regions. However, is this mechanism operating between regions at the same level in the hierarchy (e.g. left versus right PPC) or can it be a consequence of e.g. the left FEF acting on the right PPC?"

In the paragraph in line 411 I now have an improved discussion on the spatial scales on which alpha operates.

- The focus on the perceptual load theory of attention may limit the impact of the Perspective, if readers interpret the claims made as being specific to this theory. Although it is still cited reasonably

widely, I don't see perceptual load theory as a major driver of current research, particularly in cognitive and systems neuroscience research that this Perspective aims to integrate.

I have now adjusted some of the references to perceptual load theory. It is somewhat a matter of opinion. As I see it, while perceptual load might not be a major driver of current research, the theory still has a strong standing.

In contrast, I was struck that the Perspective only mentions biased competition in the final sentence before the conclusion, when this theory seems much more influential on and relevant to current neuroscience approaches to attention.

This is a point well taken. However, while I would like to think that alpha reflects the inhibitory/competitive interactions in biased competition I do not think the evidence is there. As I understand it, biased competition has mainly been studied in terms of competition between receptive fields whereas perceptual load theory has been investigated on larger spatial scales. While it might be possible to align the theories given their common elements in competitions, this is something I do not feel comfortable doing given the absence of empirical work justifying this (as far as I know).

On line 422 I write

"It would also be interesting to investigate if the principles of biased competition^{144,145} and normalization theory¹⁴⁶ in a broader sense apply to interhemispheric competition and its related to suppression by alpha oscillations."

Minor line-by-line comments

Line 22. Check reference 15. No author info etc.

Corrected thanks

Lines 178-180. Wouldn't cognitive load reduce rather than increase inhibition, following the logic and results of studies like De Fockert et al. (2001)?

This is a good point – however, it should be pointed out that in the De Fockert et al 2001 studies, digits were maintained in WM and the distractors were faces. However, in a later study, the Lavie group showed that increased WM load can result in distractor suppression where the same type of stimuli were used as memory items as well distractors:

Konstantinou N, Beal E, King JR, Lavie N. Working memory load and distraction: dissociable effects of visual maintenance and cognitive control. *Atten Percept Psychophys*. 2014 Oct;76(7):1985-97.

In the studies, we refer to regarding alpha, the items used for targets and distractors in the working memory and attention tasks are of similar type. As such, this is a complex issue and as it is peripherally relevant to the discussion on alpha and distractors, I prefer not to expand.

Line 272. "modulation" should be "modulating".

Corrected thanks

Line 370. "but SEE".

Thanks but it was left-over now deleted.

Reviewer #2 (Remarks to the Author):

The present **review** introduces a novel perspective on the indirect control mechanism governing alpha-based suppression in memory and attention, offering a fantastic and timely contribution to the field. The well-organized evidence supporting this concept aligns with my personal viewpoint. I have several suggestions to enhance the paper.

1, Concerning the term "top-down", an ongoing debate in the field of attention questions its definition (see Theeuwes, 2019, JoC and the commentaries to this paper). Generally, the argument revolves around the assertion that top-down control is intentional, effortful and goal-directed. However, the term "top-down direct control" adopted in the present **review** may not imply goal-directed control; instead, it might only refer to the direct link between alpha oscillation and suppression. Thus, using "direct and indirect control" could suffice for readers to grasp the concept.

Thanks for this insight. I have now updated the manuscript to only refer to top-down control in relation to tasks where attention is goal-directed and/or control intentional.

2, In line 35. "The increase in the magnitude of alpha oscillations with working memory load is highly robust and has been reproduced in numerous studies¹⁷⁻²⁰". While I appreciate these results, lots of experiments have shown that alpha power decreases with an increase in working memory load, (e.g., Fukuda & Woodman, 2017; Liu et al., 2022; Wang et al., 2020; for a **review** see Woodman et al., 2022). It is crucial to reconcile these findings with the current indirect control account.

This is a good point. Indeed it's a mistake that I did not address the work showing an alpha decrease with working memory load.

We now clarify on

Line 35

"It should be mentioned that the alpha power increase is observed when stimuli like letters, numbers or objects are maintained; however, if complex visuospatial information is maintained, alpha power has been reported to decrease during working memory maintenance²¹⁻²⁴. Nevertheless, the increase in the magnitude of alpha oscillations with working memory load when consonants, letters or faces are maintained is highly robust and has been reproduced in numerous studies^{19,20,25,26}."

Line 45

"The inhibition notion can also explain why alpha power does not increase when complex visual information is maintained, as in this case the maintenance likely relies on engaging a 'visuospatial sketchpad'²⁸."

Similarly, in line 180, "Also, the account of indirect modulation of distractor related alpha power can explain the absence of alpha modulation in the statistical learning studies in which distractor appearance is manipulated 71,73 (but see 65,66)". I thought the study from 66 is consistent with alpha suppression account (Woodman et al., 2022).

Indeed this part should have been clearer. We now clarify on
Line 208

“It should however be mentioned that another EEG study on statistical learning reported enhanced power in parieto-occipital alpha oscillations contralateral to frequent distractor locations⁸¹. We conclude from the EEG and MEG studies that alpha oscillations and the subsequent distractor suppression can be controlled by both direct and indirect control mechanisms. While the indirect control mechanisms appear to be dominant, further work is required to uncover in which settings indirect versus direct control mechanisms are at play. We do note that even though the increase in alpha power associated with distractor suppression is indirect, it is a secondary consequence of top-down control.”

Minor

1, Another study delves into the relationship between perceptual load and alpha, which could provide valuable insights (Wang et al., 2020).

Thanks – now added

2, The definition of alpha varies across existing literature, ranging from 8-12Hz, 8-13Hz, or 7-12Hz, etc.. Could the author provide a note/rule for future studies to follow?

We now clarify on
Line 31

“The rhythm was detected in the EEG and has a frequency range from 8-13 Hz (note that slightly other rangers are used in the literature spanning from 7 to 13 Hz).”

3, A well-defined figure describing the boundary between direct and indirect control would assist readers in understanding how alpha oscillation operates in suppression.

Thanks – I have now updated Figure 2

Fukuda, K., & Woodman, G. F. (2017). Visual working memory buffers information retrieved from visual long-term memory. *Proceedings of the National Academy of Sciences*, 201617874. <https://doi.org/10.1073/pnas.1617874114>

Liu, B., Li, X., Theeuwes, J., & Wang, B. (2022). Long-term memory retrieval bypasses working memory. *NeuroImage*, 261, 119513. <https://doi.org/10.1016/j.neuroimage.2022.119513>

Wang, S., Megla, E. E., & Woodman, G. F. (2020). Stimulus-induced Alpha Suppression Tracks the Difficulty of Attentional Selection, Not Visual Working Memory Storage. *Journal of Cognitive Neuroscience*, 1–27. https://doi.org/10.1162/jocn_a_01637

Woodman, G. F., Wang, S., Sutterer, D. W., Reinhart, R. M. G., & Fukuda, K. (2022). Alpha suppression indexes a spotlight of visual-spatial attention that can shine on both perceptual and memory representations. *Psychonomic Bulletin & Review*, 29(3), 681–698. <https://doi.org/10.3758/s13423-021-02034-4>

These reference are now included

Reviewer #3 (Remarks to the Author):

This is a really beautiful **review** article advocating the idea of posterior alpha activity reflecting a gating (instead of an inhibition) mechanism - a gating mechanism that is indirectly controlled via e.g. FEF. It is convincingly demonstrated that this gating is driven by cognitive load. I absolutely enjoyed this read, and I must admit, it also got me a bit excited about alpha activity again. Although there are quite a few similar articles out already, this **review** paper adds a novel component to the topic. The manuscript is nicely written and easy to follow, making it suitable for a broad readership.

We thank the reviewer for the kind words

I have a few minor points that the author might want to address in a revision.

1) The paragraph around line 55 describes approaches how to drive alpha activity using magnetic, electric and photic stimulation. It might be considered to also add neurofeedback-training here. I think the paper by Bagherzadeh et al. (2020) in *Neuron* would very nicely tie in here.

Excellent point. We now include and expand on
Line 74

“Another line of studies has used hemispheric neurofeedback to investigate the causal role of the alpha rhythm. Specifically, these studies allowed for modulating the hemispheric asymmetry of the alpha oscillations by neurofeedback training. The testing after training revealed biases in spatial attention in the expected direction underscoring the causal role of the alpha oscillations^{54,55} (but see⁵⁶ for a critical perspective).”

2) I am not so super convinced by the "how general is the role of alpha oscillations..." chapter. Sure, there is modulation of alpha activity in sensorimotor areas as well. But in terms of the proposed mechanism, is it really comparable with visual alpha? Hummel et al. (2002) *Brain* showed quite nicely that intentional action inhibition is associated with sensorimotor alpha activity. High alpha activity then also led to reduced cortico-spinal excitability (TMS induced MEPs). So this would actually be an example of alpha amplitude reflecting gain control in early sensory (or here motor) areas, wouldn't it. To be honest, I think the manuscript could do very well without the general role of alpha chapter.

I could not find Hummel 2002 but think this reference is relevant:

Sauseng P, Klimesch W, Gerloff C, Hummel FC. Spontaneous locally restricted EEG alpha activity determines cortical excitability in the motor cortex. *Neuropsychologia*. 2009 Jan;47(1):284-8.

I also toned down this section and stressed that the generality of alpha should be further investigated.

3) Instead, I would find it exciting to have something in the paper on how frequency specific this effect really is. There are sometimes effects very similar to those at alpha frequency reported for beta, sometimes even theta. So, in how far is this cognitive load dependent gating mechanism specific for alpha and cannot be applied to other frequencies? I am aware that this might go beyond this manuscript's scope. So I will let it to the author whether they want to include something on it.

Good point. We now added a discussion on

Line 425

“Finally, the specificity of the alpha band range deserves to be further uncovered. Often modulations of oscillations in the beta band (13 – 30 Hz) reflect engagement and disengagement in particular in the motor system; however, beta modulations are also observed in posterior regions ¹⁴⁷. While oscillations of theta oscillations are defined in the 4 – 8 Hz range in the human EEG, the hippocampal theta rhythm spans 4 – 12 Hz ¹⁴⁸. Interestingly the hippocampal theta oscillations are also a consequence of pulses of GABAergic inhibition originating from the medial septum ¹⁴⁹. This raised the possibility that theta and alpha oscillations play functionally similar roles, resulting in the hypothesis that communications between the visual system and the hippocampus might be mediated by phase-synchronization in the theta-alpha band ¹⁵⁰.”

4) around line 307 connectivity between FEF and parietal cortex and alpha is discussed. There is a paper by Sauseng et al. (2011) *Front Psychol* where FEF was TMS-ed and the effect on parietal alpha and fronto-parietal interaction was investigated during visual attention cueing. Maybe this paper could be interesting here as well.

It was a mistake to miss this reference. Now Included.

5) line 370 seems to miss a reference: ...different hemifields (but SEE)... see what?

‘But See’ now removed

6) reference 15 is not complete.

Done thanks

15th Feb 24

Dear Ole,

Your Perspective titled "Gating by alpha band inhibition revised: a case for an indirect control mechanism" has now been seen by 2 referees, whose comments appear below. In the light of their advice I am delighted to say that we are happy, in principle, to publish it in Communications Psychology under a Creative Commons 'CC BY' open access license.

We will not send your revised paper for further review if. If the revised paper is in Communications Psychology format, in accessible style and of appropriate length, we shall accept it for publication immediately. I ask you to attend to each request below in detail and complete the attached checklist.

EDITORIAL REQUESTS:

* Please check whether your manuscript contains third-party images, such as figures from the literature, stock photos, clip art or commercial satellite and map data. If any of the display items in your manuscript (figures, tables, boxes or movies) include images that are the same as, or are adaptations of, previously published images, please fill in the Third Party Rights Table, and return to us when you submit your revised manuscript. This information will enable us to obtain the necessary rights to re-use such material. If we are unable to obtain the necessary rights to use or adapt any of the material that you wish to use, we will contact you to discuss alternative options.

* Communications Psychology uses a transparent peer review system. On author request, confidential information and data can be removed from the published reviewer reports and rebuttal letters prior to publication. If you are concerned about the release of confidential data, please let us know specifically what information you would like to have removed. Please note that we cannot incorporate redactions for any other reasons.

*If you have not done so already, please alert me to any related manuscripts from your group that are under consideration or in press at other journals, or are being written up for submission to other journals (see www.nature.com/authors/editorial_policies/duplicate.html for details).

FORMATTING GUIDELINES:

You will find a complete list of formatting requirements following this link:

<https://www.nature.com/documents/commsj-style-formatting-checklist-review-perspective.pdf>

Please use the checklist to prepare your manuscript for final submission. In the following, I also highlight some issues of particular importance.

** Preface

The Abstract needs to be replaced with preface. This preface (up to 100 words; without references) should serve both as a general introduction to the topic, and as a brief, non-technical summary of your main points and their implications. It should start by outlining the background to your article (why the topic is important) and the main question you have addressed, before going on to describe

your key points, main conclusions and their general implications. Because we hope that researchers across all fields of psychology will be interested in your work, the preface should be as accessible as possible, explaining essential but specialised terms concisely. We suggest you show your preface to colleagues in other fields to uncover any problematic concepts.

** Length

The ideal length for Review Articles and Perspectives in Communications Psychology is 5,000 words. The present length is fine.

** Main text

Please use three or four section headings in the main text. These should relate to the content of the article rather than being generic. Headings should be no longer than 60 characters (including spaces) and should not use punctuation. Please do not use more than two levels of headings (section headings + subheadings).

* Please replace the subheadings that are formulated as questions with brief statements (also aiming for 60 characters or less)

** Figures

Please remove all figures from the main text and upload them individually, one figure per file. To ensure the swift processing of your paper please provide the highest quality, vector format, versions of your images (.ai, .eps, .psd) where available. Text and labelling should be in a separate layer to enable editing during the production process. If vector files are not available then please supply the figures in whichever format they were compiled in and not saved as flat .jpeg or .TIFF files. If your artwork contains any photographic images, please ensure these are at least 300 dpi.

* Figures should be simple and informative — multi-part figures are best avoided. Boxes should occupy no more than half a page in the PDF (less than 500 words) and may include a figure.

* References

References appear as superscript Arabic numerals, in order of mention. The reference list mentions references in the numerical order in which they are mentioned in the main text. If a reference is cited more than once, the same number is used throughout the text and the reference receives a single entry in the reference list.

We ask that you select the most significant 5–10% of references in your list for highlighting, and add a single sentence in bold after each of these references to describe the main result and its significance.

Only papers that have been published or accepted by a named publication should be in the reference list (preprints and citations of datasets are also permitted). Unpublished/Submitted research should not be included in the reference list; it should only be mentioned briefly and parenthetically in the main text. Note that no major arguments should rely on unpublished research.

Published conference abstracts and URLs for web sites should be cited parenthetically in the text,

not in the reference list.

Footnotes are not used.

* Competing interests

Please include a "Competing interests" statement after the References. Note that we ask authors to declare both financial and non-financial competing interests. For more details, see <https://www.nature.com/authors/policies/competing.html>. If you have no financial or non-financial competing interests, please state so: "The authors declare no competing interests."

SUBMISSION INFORMATION:

* If you wish, you may also submit a visually arresting image, together with a concise legend, for consideration as a 'Hero Image' on our homepage. The file should be 1400x400 pixels and should be uploaded as 'Related Manuscript File'. In addition to our home page, we may also use this image (with credit) in other journal-specific promotional material.

* Your paper will be accompanied by a two-sentence editor's summary, of between 250-300 characters, when it is published on our homepage. If you wish to suggest a summary, please enclose it in the cover letter of the final revision.

In order to accept your paper, we require the following:

* A cover letter describing your response to our editorial requests.

* The final version of your text as a Word or TeX/LaTeX file, with any tables prepared using the Table menu in Word or the table environment in TeX/LaTeX and using the 'track changes' feature in Word.

* Production-quality versions of all figures, supplied as separate files. Photographic images should be 300 dpi in RGB format (.jpg, TIFF or native Photoshop format) and any labels/scale bars included in a separate layer from the image. Line art, graphs and schemes should be vector format (.ai, .eps, .pdf); Adobe Illustrator files are preferred and will minimize production time. Any chemical structures or schemes contained within figures should additionally be supplied as separate Chemdraw (.cdx) files.

At acceptance, the corresponding author will be required to complete an Open Access Licence to Publish on behalf of all authors, declare that all required third party permissions have been obtained.

Please note that your paper cannot be sent for typesetting to our production team until we have received this information; **therefore, please ensure that you have this ready when submitting the final version of your manuscript.**

ORCID

Communications Psychology is committed to improving transparency in authorship. As part of our efforts in this direction, we are now requesting that all authors identified as 'corresponding author' create and link their Open Researcher and Contributor Identifier (ORCID) with their account on the Manuscript Tracking System (MTS) prior to acceptance. ORCID helps the scientific community achieve unambiguous attribution of all scholarly contributions. For more information please visit <http://www.springernature.com/orcid>

For all corresponding authors listed on the manuscript, please follow the instructions in the link below to link your ORCID to your account on our MTS before submitting the final version of the manuscript. If you do not yet have an ORCID you will be able to create one in minutes.

IMPORTANT: All authors identified as 'corresponding author' on the manuscript must follow these instructions. Non-corresponding authors do not have to link their ORCIDs but are encouraged to do so. Please note that it will not be possible to add/modify ORCIDs at proof. Thus, if they wish to have their ORCID added to the paper they must also follow the above procedure prior to acceptance.

To support ORCID's aims, we only allow a single ORCID identifier to be attached to one account. If you have any issues attaching an ORCID identifier to your MTS account, please contact the Platform Support Helpdesk.

[link redacted]

We hope to hear from you within two weeks; please let us know if the process may take longer.

Best wishes,

Marike

Marike Schiffer, PhD
Chief Editor
Communications Psychology

REVIEWERS' COMMENTS:

Reviewer #1 (Remarks to the Author):

Thank you for a responsive revision which has successfully addressed the points raised in my original review.

Line 80. "well controlled [experiments] providing evidence". Missing word?

Reviewer #2 (Remarks to the Author):

This revision has been improved significantly. I am satisfied.